# A Numerical Study on the Pilot Injection Conditions of a Marine 2-Stroke Lean-Burn Dual Fuel Engine

**Hao Guo** **, Song Zhou \*, Jiaxuan Zou and Majed Shreka**

College of Energy and Power Engineering, Harbin Engineering University, Harbin 150001, China;
guohao618@hrbeu.edu.cn (H.G.); zou_jx@hrbeu.edu.cn (J.Z.); majed.shreka@outlook.com (M.S.)
* Correspondence: songzhou@hrbeu.edu.cn; Tel.: +86-138-4506-3167

**Abstract:** The global demand for clean fuels is increasing in order to meet the requirements of the International Maritime Organization (IMO) of 0.5% global Sulphur cap and Tier III emission limits. Natural gas has begun to be popularized on liquefied natural gas (LNG) ships because of its low cost and environment friendly. In large-bore marine engines, ignition with pilot fuel in the prechamber is a good way to reduce combustion variability and extend the lean-burn limit. However, the occurrence of knock limits the increase in power. Therefore, this paper investigates the effect of pilot fuel injection conditions on performance and knocking of a marine 2-stroke low-pressure dual-fuel (LP-DF) engine. The engine simulations were performed under different pilot fuel parameters. The results showed that the average in-cylinder temperature, the average in-cylinder pressure, and the NOx emissions gradually decreased with the delay of the pilot injection timing. Furthermore, the combustion situation gradually deteriorated as the pilot injection duration increased. A shorter pilot injection duration was beneficial to reduce NOx pollutant emissions. Moreover, the number of pilot injector orifices affected the ignition of pilot fuel and the flame propagation speed inside the combustion chamber.

**Keywords:** computational fluid dynamics; two-stroke; dual-fuel engine; simulation; pre-combustion chamber

## 1. Introduction

In recent years, marine engine emission regulations have become increasingly strict [1,2]. On 1 January 2020, a 0.5% global Sulphur cap for marine fuels was implemented by the International Maritime Organization (IMO) [3]. The North Sea, the Baltic Sea, and the North American coast have been set as emission control zones (ECAs). In order to meet the IMO requirements of the Tier III emission standard and the 0.5% global Sulphur cap, many marine diesel engine manufactures are looking for engines with lower emissions and higher combustion efficiency [4]. Natural gas (NG) has gradually become a promising engine fuel solution in order to meet these needs [5].

Compared with the high cost of low Sulphur fuel oil (LSFO), exhaust gas recirculation (EGR), and selective catalytic reduction (SCR), natural gas engines have good development prospects because of its low cost [6]. The main component of natural gas is methane ($CH_4$), which has the advantages of low heat value, good explosion resistance, abundant reserves, and good emission performance [7,8]. However, natural gas requires higher ignition energy, which is often fulfilled by employing spark plug ignition or a small amount of pilot diesel. For large marine vessels, the dual-fuel low-speed engine can flexibly switch between diesel mode and gas mode, which solves the problem of unstable gas supply [9]. The dual-fuel engines use natural gas as fuel and their thermal efficiency is similar to that of diesel engines, which means low economic costs and emissions.

Winterthur Gas and Diesel (WinGD) manufactured the first 2-stroke low-pressure dual-fuel (LP-DF) engine. In the WinGD RT-Flex50DF engine, the natural gas is injected into the cylinder at a

lower pressure after the piston passes through the scavenging port, which leads to mixing natural gas with air [10]. When the piston moves to the top dead center (TDC), a small amount of pilot oil (1%) is sprayed into the cylinder. The ignition energy of the pilot fuel is used to ignite the $CH_4$-air mixture in pre-combustion chamber (PCC), burning the fuel in the main chamber and completing the power stroke. Compared to conventional diesel engines, LP-DF engines significantly reduce emissions while ensuring high combustion efficiency, thus easily meeting the Tier III emission standards without any after-treatment system [11]. LP-DF engines can also switch freely between gas mode and diesel mode according to different operation requirements. LP-DF technology can achieve high-quality premixed lean combustion—making the engine power and thermal efficiency close to traditional low-speed engines.

The advantages of the low-pressure premixed dual-fuel engines are significant but engine loads and maximum output torque are limited due to knocking [12]. In order to solve these problems, the heat release, the flame propagation, the turbulence distribution, and the end-gas conditions in the LP-DF engine cylinder must be investigated. The computational fluid dynamics (CFD) tool is used to analyze the engine combustion situation and in-cylinder flow [13,14]. Hockett et al. used a reduced chemical kinetic mechanism consisting of 141 species and 709 reactions to simulate the combustion of natural diesel and gas fuels in a dual-fuel engine [15,16]. CFD simulation model using the reduced mechanism achieved an excellent agreement with the experiment data at different loads and injection timings. They found that the model can predict knock phasing.

In the design of a lean-burn dual-fuel engine with Otto cycle, it is important to understand the flame propagation and abnormal combustion. Filip et al., used a quasi-dimensional combustion model to predict 2-stroke LP-DF engine performance. An extensive study related to both turbulent and laminar flame velocity was accomplished and a quasi-dimensional simulation approach was proposed [17]. It was found that the ignition delay is determined for both gaseous and pilot fuel. Furthermore, the combustion tended to be triggered by self-ignition at advanced pilot injection timing and at elevated intake temperature.

Duan et al. [18,19] studied the effects of the controlling strategies on the auto-ignition timing and combustion phase in the homogeneous charge compression ignition (HCCI) engine. Controlling strategies (such as fuel management, EGR, homogeneous charge preparation) can be used to control the compressed gas temperature, the pressure, and the mixture in-cylinder distribution at the end of the compression stroke so that the fuel could be auto-ignited at the desired crank angle, and thereby obtaining optimal heat release rate and combustion phase.

The diffusion combustion of the pilot fuel and the propagation of premixed gas flame in the LP-DF engine were difficult to describe [20]. Amin et al. studied the combustion process in an LP-DF engine by coupling detailed chemical kinetics and CFD simulation, and found that more pilot diesel caused ignition delay and increased peak in-cylinder pressure [21]. Mavrelos et al. focused on the comprehensive investigation of a large 2-stroke lean-burn dual-fuel engine by using GT-Suite software [22]. He found that the pilot injection timing was advanced to avoid knocking and the lean-burn engine compression ratio was lower. This led to an earlier start of combustion, shorter combustion duration, and higher peak pressure in the gas mode compared to the diesel mode operation.

The marine LP-DF engine is different from the traditional diesel engine in the processes of air exchange, mixture formation, ignition, and flame propagation [23,24]. So far, the combustion processes and mixing in the lean-burn dual-fuel engine have not been fully understood. Liu et al. [25] conducted a CFD simulation on the flow motion and combustion process in the main chamber and the prechamber for a low-speed dual-fuel engine. He found that the enhancement of the in-cylinder swirl can prevent the spontaneous combustion of the end-gas before the flame reach and reduce the possibility of knock.

Pilot injection conditions are an important parameter of lean-burn dual-fuel engines and it has a significant effect on the dual-fuel engine performance and knocking. Liu et al., investigated the effect of the $CH_4$ equivalence ratio and pilot fuel mass on the ignition/extinction of the fuel/air mixture and the in-cylinder tendency of knocking [26]. He found that the increase in the $CH_4$ equivalence ratio can

improve the LP-DF engine power but it also induced promotes NOx emissions and pressure oscillation. Moreover, the pressure oscillation increased with the increase in the pilot fuel amount.

In a marine 2-stroke lean-burn dual-fuel engine, the occurrence of knock limits the increase in power [27]. Therefore, the aim of this study is to improve the dual-fuel engine performance and to reduce the occurrence of knocking by optimizing pilot injection conditions. A three-dimensional (3D) model of the 2-stroke LP-DF engine was developed and validated with the help of CFD simulation software (Section 2). Then, the effects of pilot fuel injection conditions in PCC on LP-DF engine emissions and performance were studied from three aspects: injector orifices, pilot fuel SOI timing, and injection duration (Section 3). The main simulation results and conclusions were summarized at last (Section 4).

## 2. Model Description

### 2.1. Engine Dimensions

WinGD RT-Flex50DF engine is designed to meet emission requirements far below the IMO Tier III limits—utilizing a highly efficient, low cost, and reliable low-pressure gas supply system. This engine can achieve smokeless combustion at all running speeds under gas operation mode. More details about the main parameters of RT-Flex50DF engine are provided in Table 1.

**Table 1.** Winterthur Gas and Diesel (WinGD) 6RT-Flex50DF engine dimensions.

| Parameter | Value |
|---|---|
| Bore/Stroke | 500/2050 mm |
| Engine Speed | 124 r/min |
| IMEP | 17.3 bar (at R1) |
| Engine Output | 8640 kW |
| Compression Ratio | 12 |
| Number of PCC | 2 |
| Brake Specific Pilot Fuel Consumption (BSPC) (GAS Mode at R1) | 1.8 g/kWh |
| Brake Specific Gas Consumption (BSGC) (GAS Mode at R1) | 142.7 g/kWh |

As demonstrated in Figure 1, a GT-Suite model of the WinGD 6RT-Flex50DF engine was developed for 1D calculation [28]. The 1D model was built to simulate the working processes of the dual-fuel engine, which can provide the required initial conditions and boundary conditions for the CFD model. The heat release rate data in the GT model was from the engine experiment under 100% load, so the validated 1D model can better support the 3D CFD analysis.

The schematic diagram in Figure 2 shows an overview of the involved calculation tools for LP-DF engine optimization analysis. Firstly, the GT-Suite model can quickly and accurately calculate the scavenging pressure and the initial in-cylinder temperature, so the 1D simulation results can better support the CONVERGE software 3D simulation to obtain an accurate compression pressure curve. Secondly, the CAD software is used for 3D chamber geometry modeling and the gas nozzles and PCC meshes are refined to better simulate the turbulence and fuel injection. Finally, a stable and efficient working range of 6RT-Flex50DF engine is studied based on the flame initiation and end gas conditions.

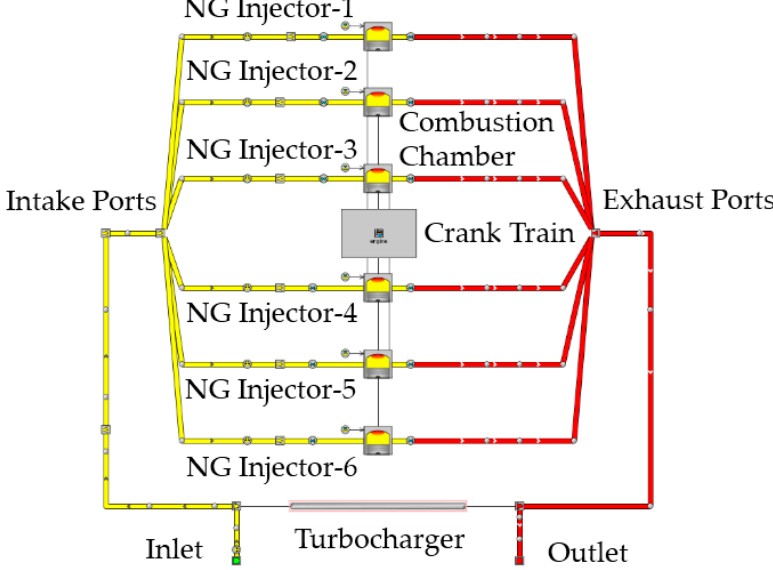

**Figure 1.** Simulation model of a 6RT-Flex50DF engine.

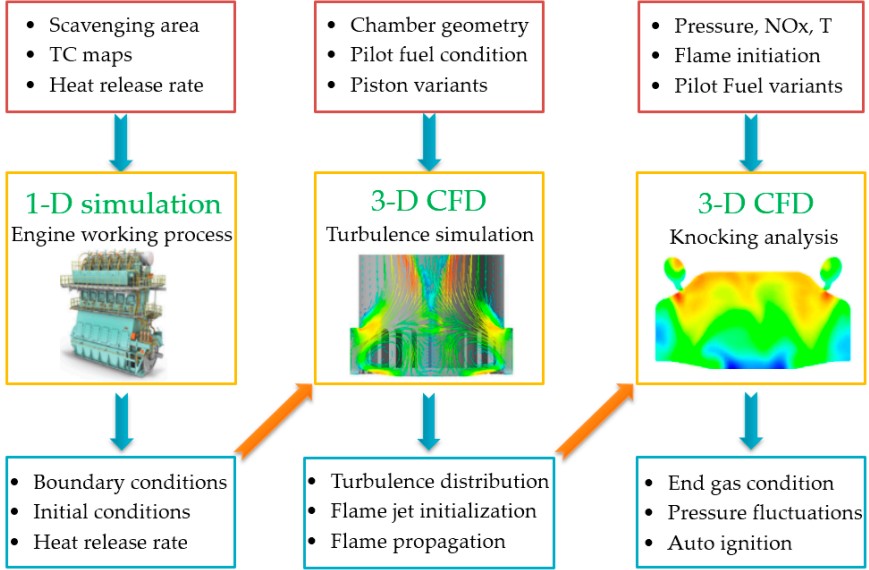

**Figure 2.** Schematic diagram of a low-pressure dual-fuel (LP-DF) engine simulation.

### 2.2. Multidimensional CFD Model

From the cylinder work of an LP-DF engine in Figure 3, natural gas is injected into the cylinder during the compression stroke. In order to make the mixture more uniform, two natural gas nozzles are symmetrically arranged in the lower position of the RT-Flex50DF engine cylinder. The pilot diesel is injected into the PCC and the lean mixture in PCC is ignited close to the compression TDC. Meanwhile, a high-speed hot jet flame formed leads to the ignition of the main combustion chamber (MCC). Moreover, the LP-DF engine starts the power stroke after the flame spreads throughout the chamber.

Figure 4 illustrates the arrangements of PCC and MCC in the RT-Flex50DF engine. The two small ellipsoidal PCC are arranged on the top of the main combustion chamber. The CFD model was secondary performed embedding of the gas nozzles, the PCC, and the spray position to achieve higher computation accuracy. After the refinement of the DF engine mesh, the maximum number of simulation cells was 628,270 at the bottom dead center (BDC). The simulation range was from 110 °crank angle (CA) to 470 °CA.

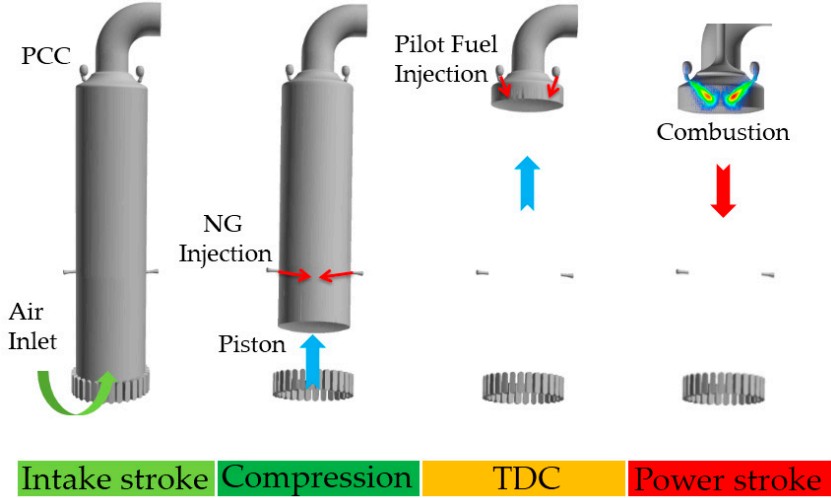

**Figure 3.** Cylinder work of a LP-DF engine (PCC: Pre-combustion chamber, NG: Natural gas, TDC: Top dead center).

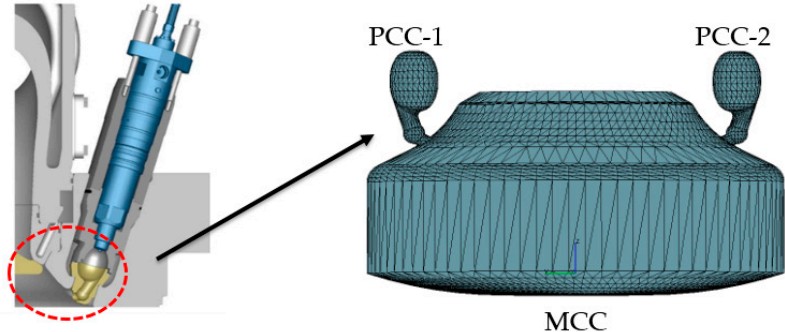

**Figure 4.** PCC and main combustion chamber (MCC) arrangements from a LP-DF engine.

The KH-RT model in the spray breakup model and the O'Rourke collision model were used for the CFD calculation. The standard K-ε turbulence model and the detailed chemistry combustion model were used in the simulation. Moreover, the extended Zeldovich NOx emission model was chosen as the NOx emission model.

The atmospheric pressure in the engine experiment was 101.7 kPa and the scavenging air temperature was 302.1 K. Furthermore, the lower calorific value of the natural gas fuel was 47.64 MJ/kg and the natural gas contained 97.07 mol% $CH_4$ and 2.93% other gases (0.24 mol% ethane, 0.01 mol% propane, etc.). The excess air ratio of the lean-burn dual-fuel engine was close to 2.2. Moreover, the oxygen content in the flue gas under 25%, 50%, 75%, and 100% engine loads were 13.59%, 14.21%, 14.26%, and 13.91%, respectively. In addition, heated flame ionization HC detector (HFID) and chemiluminescent NOx analyzers were used in the experiment.

Referring to the mean in-cylinder pressure in Figure 5a, the error between the experiment and the CFD calculation results is less than 3.6%. As shown in Figure 5b, the NOx emissions predictions are at a very good level over the entire DF engine load range. From all the above, the comparison between measured data and calculation results gives good agreements in terms of the mean in-cylinder pressure and the NOx emissions, which meets the accuracy requirements of the CFD simulation.

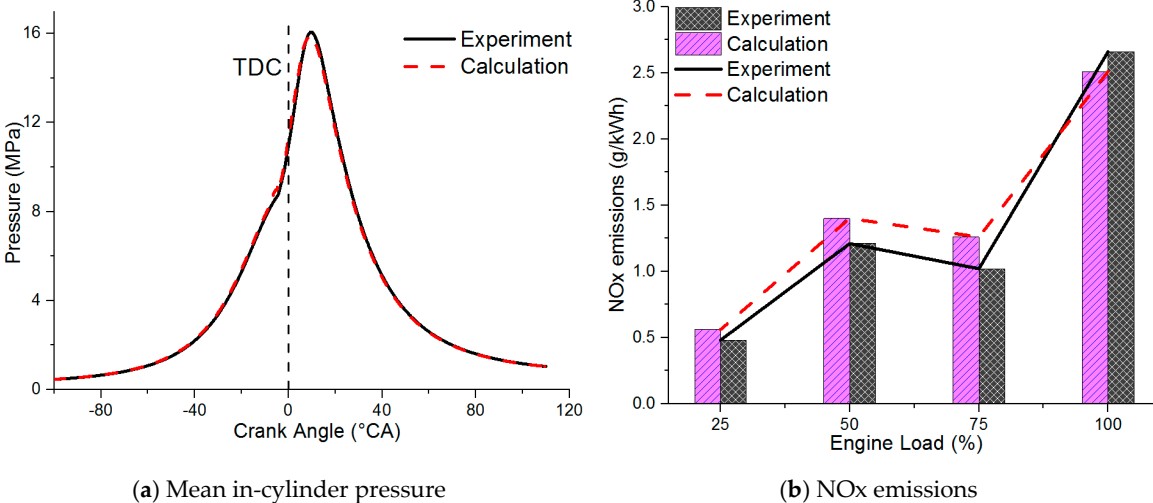

(**a**) Mean in-cylinder pressure            (**b**) NOx emissions

**Figure 5.** Simulation data (dashed) of mean in-cylinder pressure and NOx emissions against experimental data (solid).

## 3. Results and Discussion

### 3.1. Pilot Injector Orifices Variation

Keeping the conditions of the prechamber geometry and the engine operating conditions unchanged, three comparison cases of pilot injector orifices of one, two, and four were selected for simulation. In the case of one orifice and two orifices, it simulates that the nozzles are clogged under extreme engine conditions.

Figure 6 shows the temperature distribution in the combustion chamber with different numbers of pilot injector orifices. It can be seen that different numbers of pilot injectors have a significant effect on the flame propagation speed and the temperature distribution in the prechamber. At −4 °CA before the top dead center (bTDC), the flame starts entering the main combustion chamber when the pilot injector is four orifices. Nevertheless, the flame penetrates half of the main combustion chamber when the pilot injector is 1 orifice, and the flame center temperature is obviously higher. At −3 °CA bTDC, the flame basically spreads to the bottom of the combustion chamber at one injector orifice and the flame temperature is significantly higher.

By comparing the temperature contours in Figure 6, the time when the flame spreads throughout the entire combustion chamber with two injector orifices and four injector orifices is significantly later than the time under single injector orifice. In addition, the temperature distribution in the chamber is not uniform and the temperature is low when the number of injector orifices is four. As the number of pilot injector orifices increases, the flame propagation speed in the combustion chamber slows down significantly and the temperature distribution in the prechamber is more uneven. This is due to the small volume of the prechamber. Under the same injection pressure of pilot, increasing the number of injector orifices will lead to the collision between the pilot and the wall of PCC, which will affect the evaporation and ignition of the pilot fuel.

Figure 7 shows the distribution of the velocity vector field with different numbers of pilot injector orifices at 2 °CA after pilot fuel injection. It can be seen that the velocity of the flame jet center under the single pilot injector orifice is the largest. From Figures 6 and 7, the speed of the flame jet decreases with the increase of injector orifices number, which indicates that different injector orifices will affect the combustion of the pilot fuel in the PCC. Furthermore, the flame jet entering the main combustion chamber greatly increases the turbulence in the cylinder and promotes the uniform mixing of natural gas and air, which can further promote the natural gas combustion.

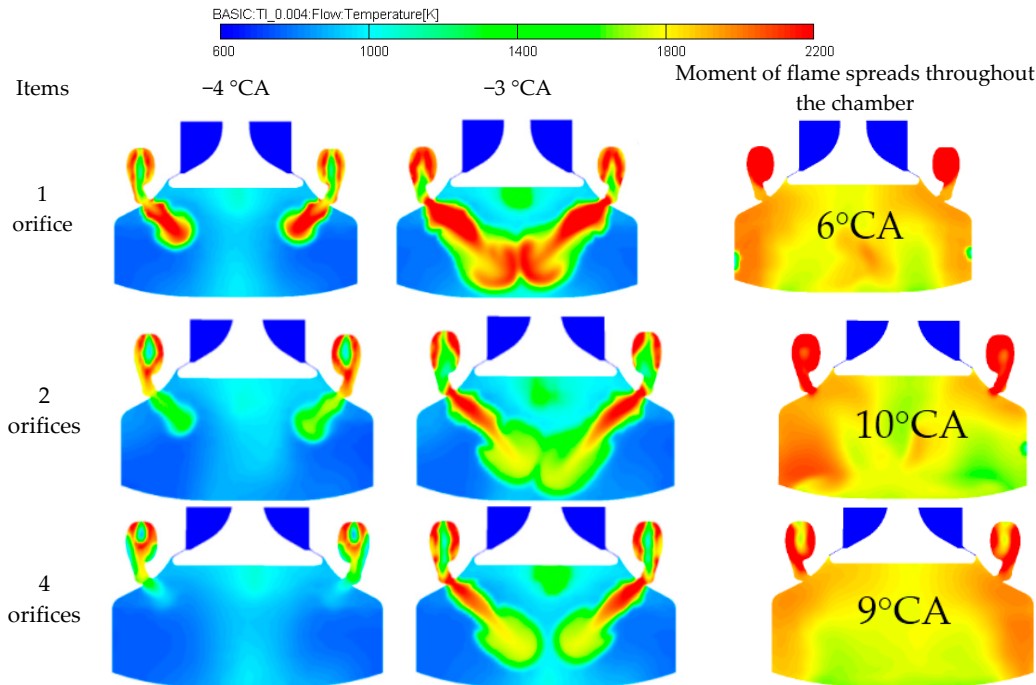

**Figure 6.** Flame distribution in the combustion chamber.

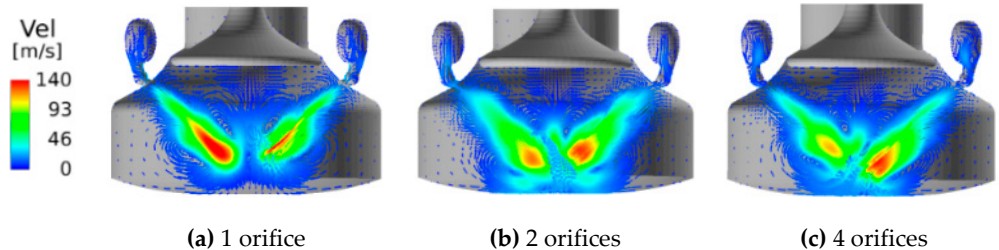

**(a)** 1 orifice　　　　　　**(b)** 2 orifices　　　　　　**(c)** 4 orifices

**Figure 7.** Flame velocity vector distribution.

As shown in Figure 8, the pressure in the cylinder gradually decreases as the number of pilot injector orifices increases from one to four but the decreasing trend becomes weaker and the time when the pressure peak appears is gradually delayed. Figure 8 shows the rate of heat release (ROHR) under different pilot injector orifices numbers. It can be seen that the heat release rate of the mixture decreases and the heat release duration increases as the number of pilot injector orifices increases. When the number of nozzle orifices is one, the heat release duration is shorter and the heat release is more rapid, which is beneficial to the rapid combustion of the natural gas–air mixture.

There are two pressure peaks, as can be seen from the mean in-cylinder pressure with different injector orifices in Figure 9. The first pressure peak is caused by the combustion of pilot diesel and occurs before TDC. The second pressure peak is due to the flame jet entering the MCC, which ignites the lean natural gas–air mixture. Furthermore, the pressure peaks caused by the pilot fuel ignition are relatively close when the number of pilot injector orifices is two and four. However, the peak pressure is higher than the other two cases and increases significantly when the engine is under a single orifice injection. The combustion in PCC under a single orifice injection is faster and a shorter combustion duration can be observed.

Figure 10 depicts the mean temperature variation in the PCC and the MCC with different injector orifices. For the PCC and the MCC, it can be seen that the temperature in the chamber gradually decreases as the number of pilot injector orifices increases and the peak temperature occurs later. When the number of injector orifices is two and four, the changes in average temperature in MCC are almost the same. In addition, the combustion temperature decline range in PCC and MCC are

quite different under different injector orifices. The temperature fluctuation in the PCC is much higher than that of the main combustion chamber, the PCC average peak temperature dropped by 293 K while the MCC average peak temperature dropped by 48.5 K. The main reason for PCC with a higher temperature is that there is more residual exhaust gas in PCC and poor heat dissipation.

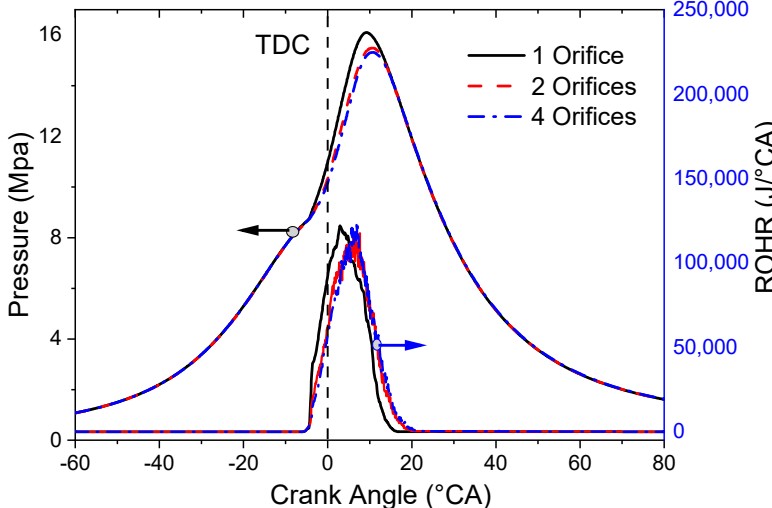

**Figure 8.** In-cylinder pressure and heat release rate with different injector orifices.

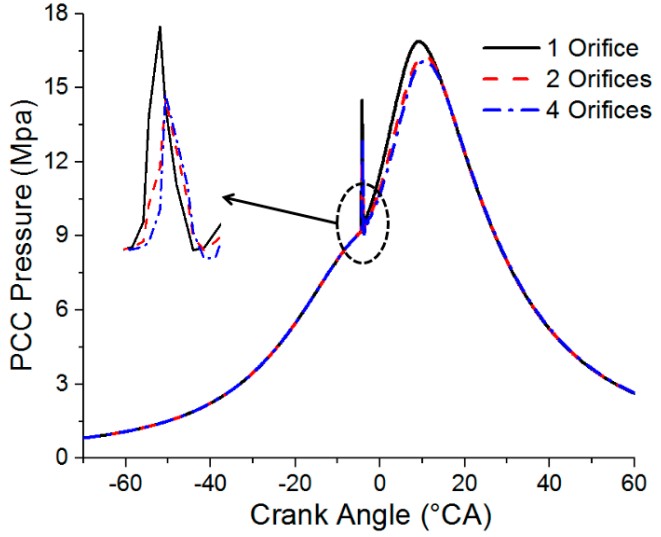

**Figure 9.** Mean in-cylinder pressure with different injector orifices.

Figure 11 shows the NOx and hydrocarbon (HC) emissions with different injector orifice numbers. The NOx production amount gradually increases and HC emissions decrease with the increase in the number of pilot injector orifices. Due to the difference in the temperature of the combustion chamber, the final NOx pollutant emissions are different in each case. Although the mean temperature under a single injector orifice is higher, the combustion duration is shorter with lower NOx emissions. Among the three simulation cases, engine performance and NOx emission characteristics are both better for the single injector orifice case. Furthermore, $CH_4$ accounts for the majority of the HC emissions and had the highest concentration. Methane slip and unburned HC are the main causes of HC emissions from LP-DF engines. Since only the number of injector orifices is changed in the simulation cases, the mass of the direct methane slip remains unchanged. The results show that the combustion duration is longer and the combustion is more sufficient under four injector orifices, which leads to the reduction of unburned HC.

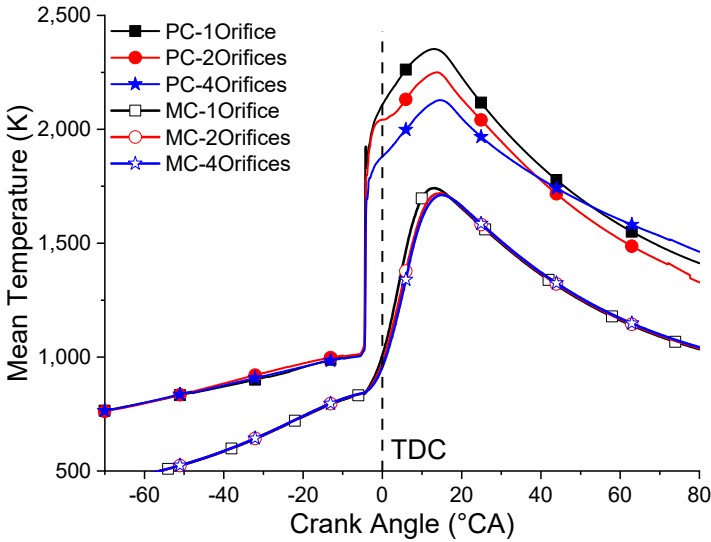

**Figure 10.** Mean in-cylinder temperature with different injector orifices.

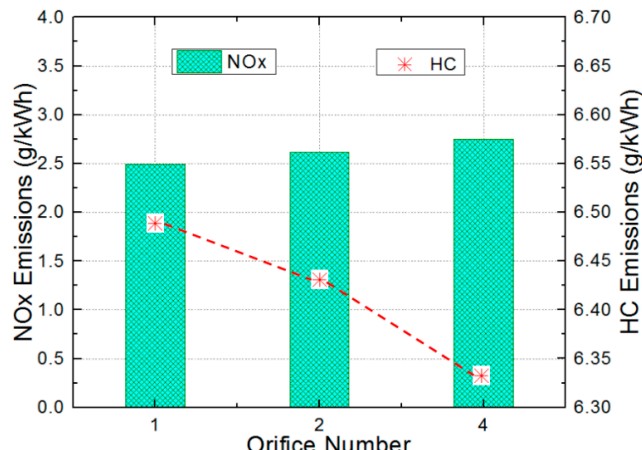

**Figure 11.** NOx and HC emissions with different injector orifices.

## *3.2. Pilot SOI Timing Variation*

### 3.2.1. Influence on Combustion

The Start of Injection (SOI) timing of pilot fuel was changed and four different pilot SOI timing durations including −8 °CA, −5 °CA, −3 °CA, and 0 °CA (TDC) were chosen for simulation to study the effects of pilot SOI timing variation on the performance and emissions of the LP-DF engine. The other parameters of the LP-DF engine were kept unchanged.

Figure 12 shows the temperature distribution in the combustion chamber at different pilot SOI timings. Comparing contours of temperature distribution, it can be seen that different pilot SOI timings have no obvious effect on the propagation of flame. However, studying the cylinder center area under the exhaust valve, with the delay of pilot injection, the temperature in the cylinder center area is higher, which may cause spontaneous combustion of some mixtures. The duration from SOI timing to the moment of flame spreading through the chamber is 11 °CA when the pilot SOI timing is −8 °CA, −5 °CA, and −3 °CA bTDC. When the pilot SOI timing is 0 °CA, the combustion duration is 13 °CA. The results show that changing the timing of pilot injection has a relatively small effect on the flame propagation speed but affects the ignition time of the mixture and in-cylinder temperature distribution.

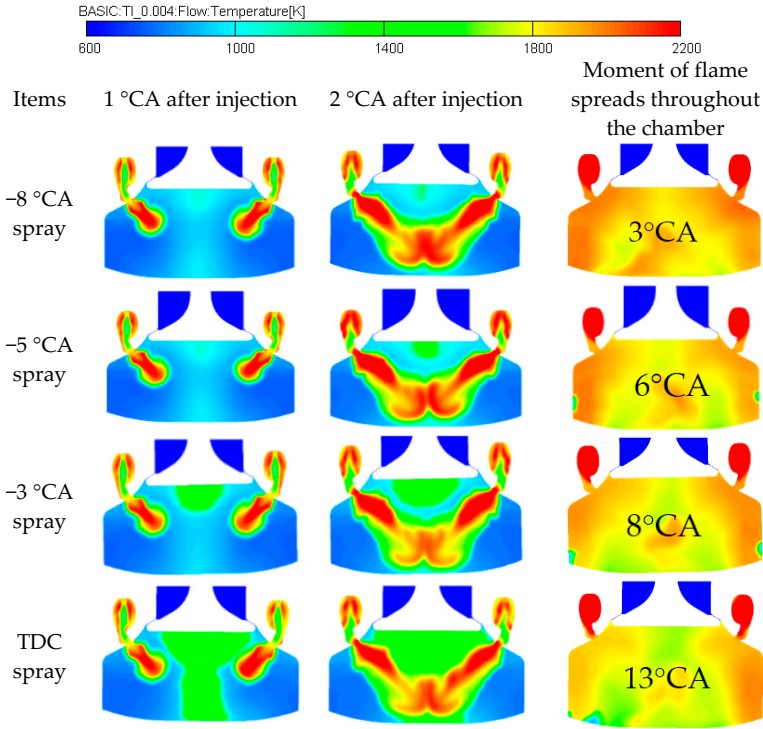

**Figure 12.** Temperature distribution in the combustion chamber.

Figure 13 denotes the distribution of the velocity vector under different pilot fuel SOI timings at the same time of 2 °CA after pilot fuel injection. It can be seen that with the change of pilot fuel injection timing, the overall flame propagation velocity does not change much and the jet flame speed under 0 °CA SOI timings is the lowest.

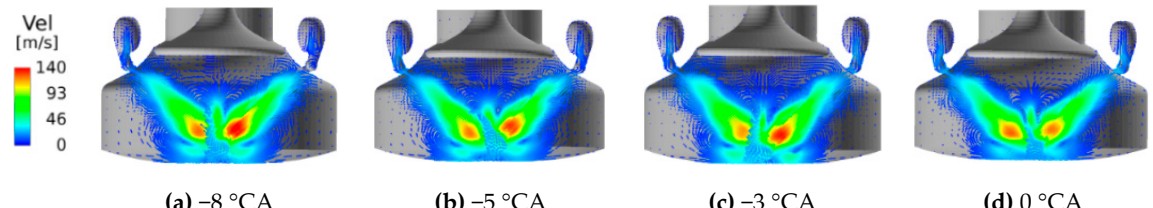

**(a)** −8 °CA      **(b)** −5 °CA      **(c)** −3 °CA      **(d)** 0 °CA

**Figure 13.** Flame velocity vector distribution at 2 °CA after pilot injection.

3.2.2. Influence on Performance

Figure 14 shows the variation of cylinder pressure with different pilot fuel SOI timings. It can be clearly seen that the in-cylinder pressure gradually decreases and the pressure peak is also delayed with the delay of the pilot SOI timing. The highest in-cylinder pressure occurs at the pilot SOI timing of −8 °CA. The ROHR curves are also depicted in Figure 14 under different pilot SOI timing. It can be seen that as the timing of pilot injection is delayed, the peak heat release rate gradually decreases and the heat release duration of the mixture gradually increases. As a result, the mixture combustion situation gradually deteriorates with the delay of pilot injection.

Figure 15 illustrates the pressure change curves in PCC at different pilot injection timings. With the delay of the pilot SOI timing, the first peak pressure in PCC caused by the pilot fuel ignition first increases and then decreases. The first pressure peak is the largest when the pilot SOI timing is −5 °CA. The maximum pressure of the second peak in the pre-combustion chamber gradually decreases with the delay of pilot injection time. In addition, the occurrence of peak pressure in PCC is gradually delayed with the delay of injection time.

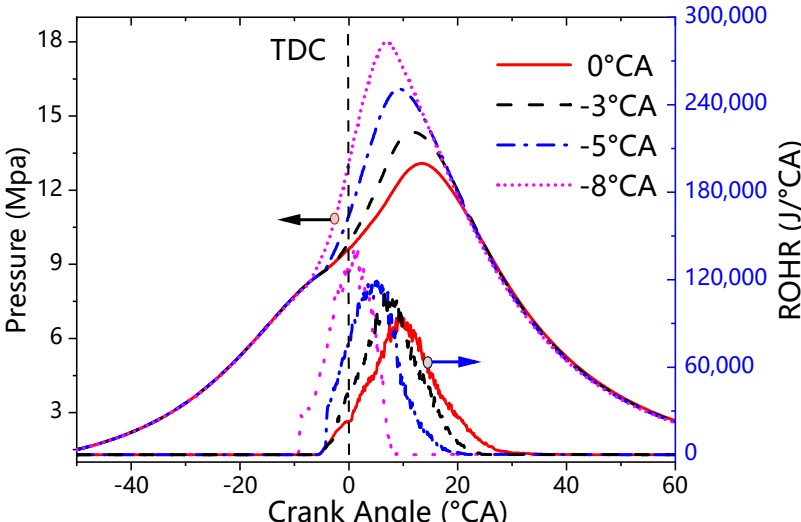

**Figure 14.** Pressure and heat release rate at different pilot start of injection (SOI) timing.

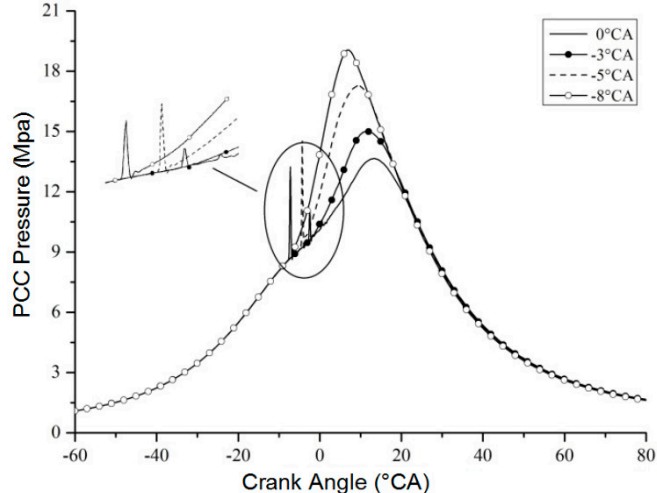

**Figure 15.** Mean in-cylinder PCC pressure at different pilot SOI timing.

Figure 16 depicts the mean temperature variation in PCC and MCC under different pilot SOI timing. Similar to the law of pressure change in the cylinder, the average temperature in PCC and MCC both gradually decreases with the delay of pilot fuel injection. When the SOI timing is in the rang from −8 °CA to 0 °CA, the peak temperature in MCC is delayed from 7 °CA to 21 °CA, whilst the peak temperature in PCC is delayed from 13 °CA to 18 °CA. The temperature fluctuation in PCC is also higher than that of MCC, the PCC peak temperature dropped by 298 K while the MCC average peak temperature dropped by 192 K.

Figure 17 shows the NOx and HC emissions under different pilot SOI timings. The emissions curves increase first and then decrease due to the oxidation of NO in the later combustion stage. It can be seen that the NOx emissions gradually decrease as the pilot fuel injection timing delays but the changes in NOx emissions are very small when the injection timing is in the range from −5 °CA to 0 °CA. When the pilot injection timing is −8 °CA, the NOx production is significantly higher than the other three cases. This is because the average in-cylinder temperature at −8 °CA SOI timing is the highest and the amount of NOx products is the largest with enough oxygen. It can be seen from Figure 17 that HC emissions decrease as the timing of pilot injection delays but the reduction is not obvious.

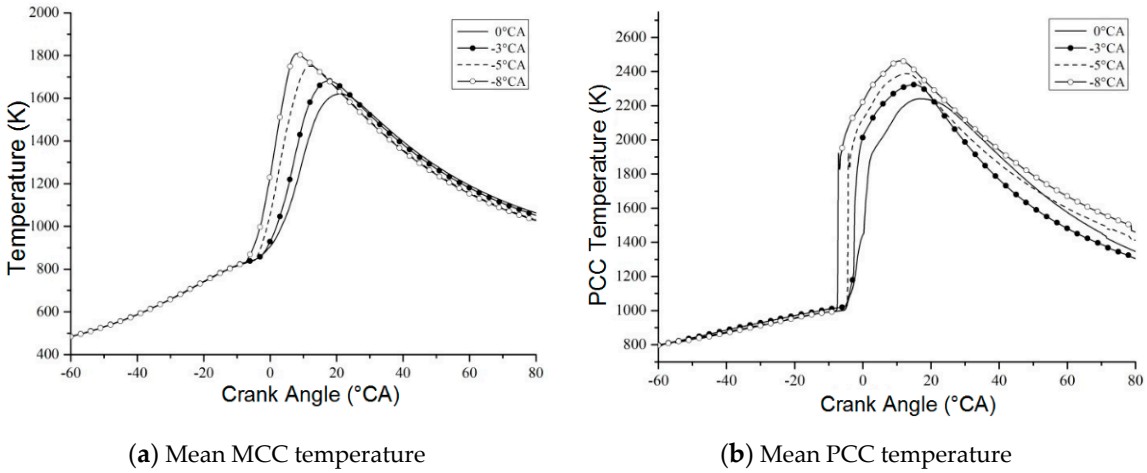

(**a**) Mean MCC temperature          (**b**) Mean PCC temperature

**Figure 16.** Mean temperature under different pilot SOI timing.

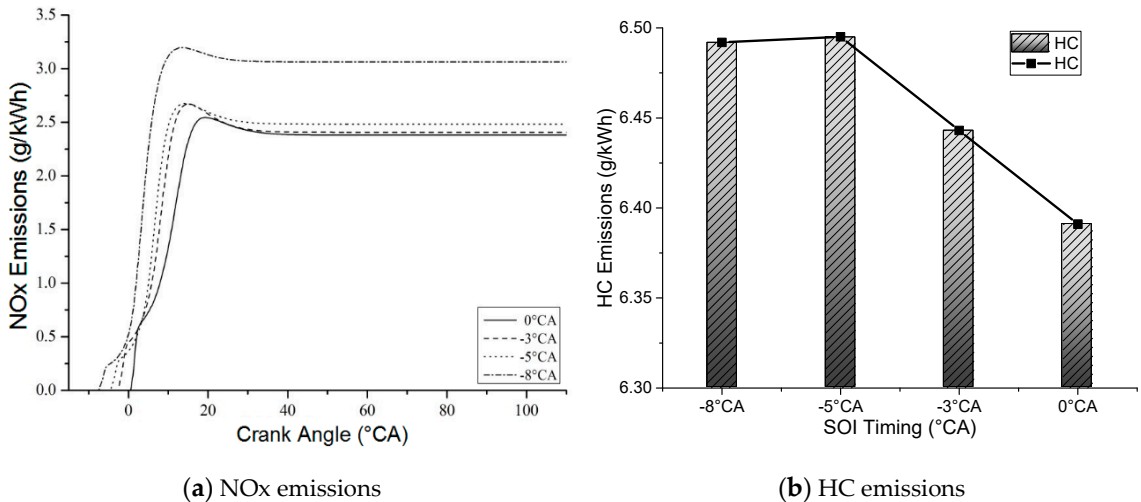

(**a**) NOx emissions          (**b**) HC emissions

**Figure 17.** NOx and HC emissions under different pilot SOI timing.

### 3.3. Pilot Injection Duration Variation

#### 3.3.1. Influence on Combustion

In order to study the effects of pilot injection duration on the performance and emissions of the LP-DF engine, four different injection durations including 1 °CA, 3 °CA, 5 °CA, and 10 °CA were used. The duration of the pilot was different under different injection pressure.

Figure 18 shows the temperature distribution in the combustion chamber under different pilot injection durations. It can be seen that as the duration of pilot becomes longer, the ignition time of pilot fuel is delayed. When the pilot injection duration is 1 °CA, the pilot fuel ignition time is 0.5 °CA after injection. When the pilot injection duration increases to 10 °CA, the pilot fuel ignition time is 3 °CA after injection, and the ignition delay period is extended with a slower injection. Comparing the four cases of temperature contours, it can be seen that the temperature of the pilot fuel jet flame gradually decreases when the pilot injection duration gradually increases, and the temperature distribution in the prechamber changes accordingly. Moreover, the average temperature in MCC has risen significantly with faster injection.

When the pilot injection duration is 1 °CA, the flame spreads throughout the chamber at 6 °CA aTDC, and combustion propagation has lasted for 10.5 °CA. Furthermore, the flame spreads throughout the chamber is 9 °CA aTDC when the pilot injection duration increases to 3 °CA and the combustion

propagation lasts has lasted about 13 °CA. The flame propagation situation is relatively close when the pilot injection durations are 5 °CA and 10 °CA, in which the combustion in the two cases is weak.

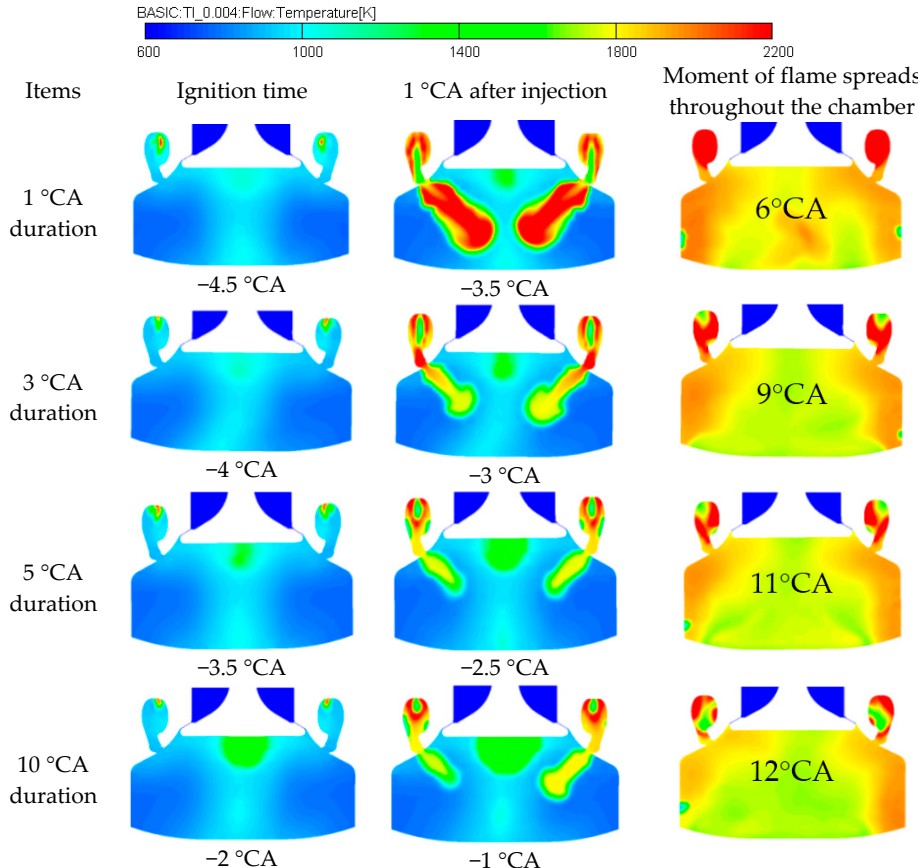

**Figure 18.** Temperature distribution in the combustion chamber.

Figure 19 depicts the distribution of the velocity vector under different pilot injection durations at 2 °CA after the pilot injection. From Figure 19, the flame propagation speed is greatly affected by the pilot injection. When the pilot injection duration is 10 °CA, the flame propagation speed is significantly lower than that of 1 °CA. The rapid heat release of pilot oil can cause a stronger flame jet, thereby promoting the turbulent combustion. Through the above analysis, the injection duration affects the ignition delay of pilot fuel and the speed of flame propagation and has a certain impact on the temperature distribution in the prechamber.

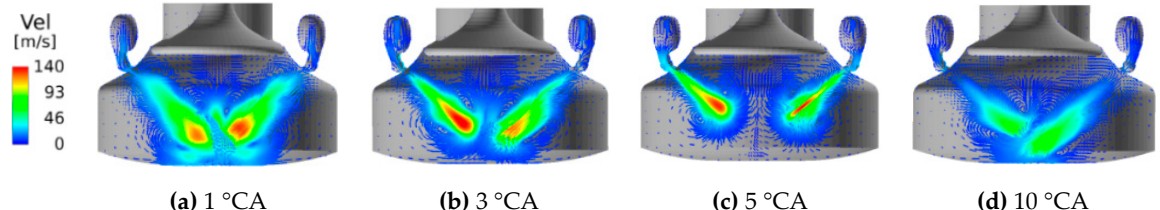

**(a)** 1 °CA       **(b)** 3 °CA       **(c)** 5 °CA       **(d)** 10 °CA

**Figure 19.** Flame velocity vector distribution.

### 3.3.2. Influence on Performance

Figure 20 shows the simulation mean pressure in the cylinder and the heat release rate with different pilot injection durations. When the pilot injection duration is 1 °CA, the in-cylinder pressure is significantly higher than the pressure obtained by the other three cases. Furthermore, the ROHR is also higher than the other cases under a pilot injection duration of 1 °CA. As the duration of pilot

injection increases, the appearance of the peak in-cylinder pressure is gradually delayed. When the pilot injection duration is 1 °CA, the heat release rate is significantly faster and the peak value is larger. Compared with the other three pilot injection durations, the mixture heat release time is the shortest under 1 °CA pilot injection duration. Moreover, the mixture burns faster and the combustion situation is relatively better under fast pilot injection. The heat release rate of the mixture in the other three cases has similar changes. As the pilot injection duration increases, the peak heat release rate gradually decreases and the downward trend slows down. This means that as the duration of pilot injection increases, the combustion gradually deteriorates with weak turbulent kinetic energy in the cylinder. With the increase of the pilot injection duration, the combustion becomes worse, the heat release duration of the mixture increases, and the in-cylinder pressure gradually decreases—resulting in a decrease in the engine performance.

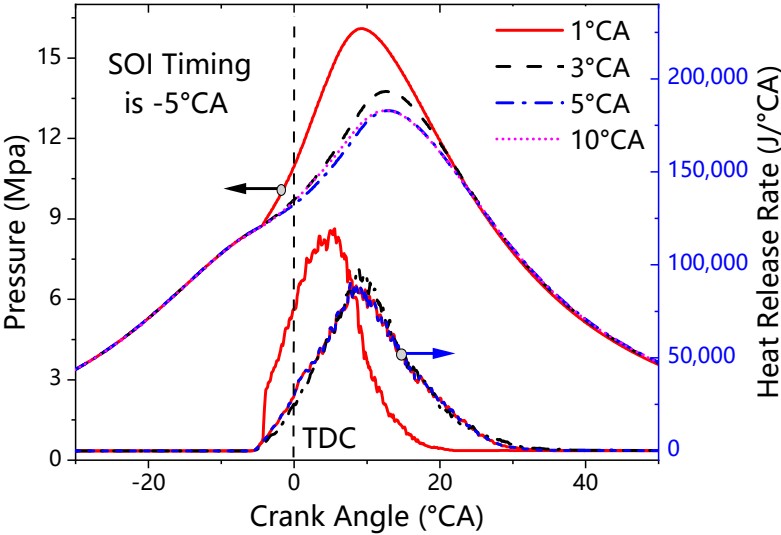

**Figure 20.** Mean pressure and rate of heat release at different pilot injection durations.

Figure 21 denotes the variation of the average temperature in the cylinder with different pilot injection durations. With the increase of the pilot injection duration, the in-cylinder temperature gradually decreases and the time when the temperature peak occurs is gradually delayed. It can be seen from Figure 21a that when the pilot injection durations are 5 °CA and 10 °CA, the changes in average temperature in MCC are basically the same. With the change of pilot injection duration, the temperature variation in PCC is very obvious and the peak temperature in the prechamber gradually decreases with the increase of pilot injection duration.

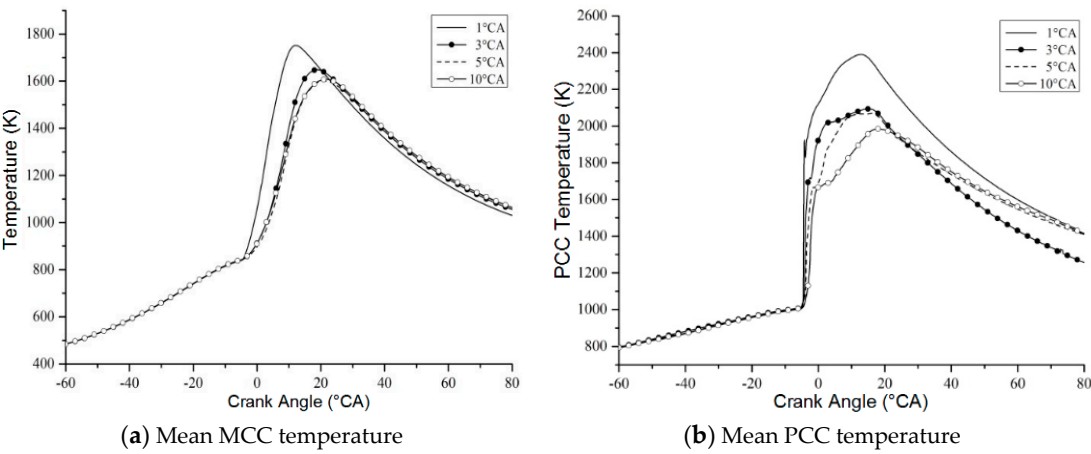

(**a**) Mean MCC temperature          (**b**) Mean PCC temperature

**Figure 21.** Mean temperature under different pilot injection durations.

Figure 22a depicts the variation of NOx emissions under different pilot injection durations. In the early stage of combustion, the amount of NOx emissions decreases with the increase of pilot injection duration. When the pilot injection duration is 1 °CA, the NOx emissions trend rises first and then decreases but this trend is not obvious under the other three cases. This is because the first NOx emissions peak is mainly caused by the combustion of pilot at the beginning of the combustion process, which is obviously affected by the average temperature in PCC. Finally, at the end of combustion, NOx emissions under 10 °CA injection duration are the highest while NOx emissions under 1 °CA injection duration are the least. This reason needs to be explained by combining the average temperature in Figure 21a and the ROHR in Figure 20, It is found that when the pilot injection duration is 1 °CA, the average MCC temperature is the highest. The generation of NOx emissions is greatly affected by the temperature in the cylinder and the duration of high temperatures. However, the heat release duration of the mixture is shorter than the other three cases and the smaller duration of high temperature reduces the formation of NOx emissions. It can be seen from Figure 22b that the amount of HC generated is highest when the pilot injection duration is 1 °CA. With the increase of the pilot injection duration, HC emissions decrease first and then increase. When the pilot injection duration is 3 °CA, HC emissions are the smallest.

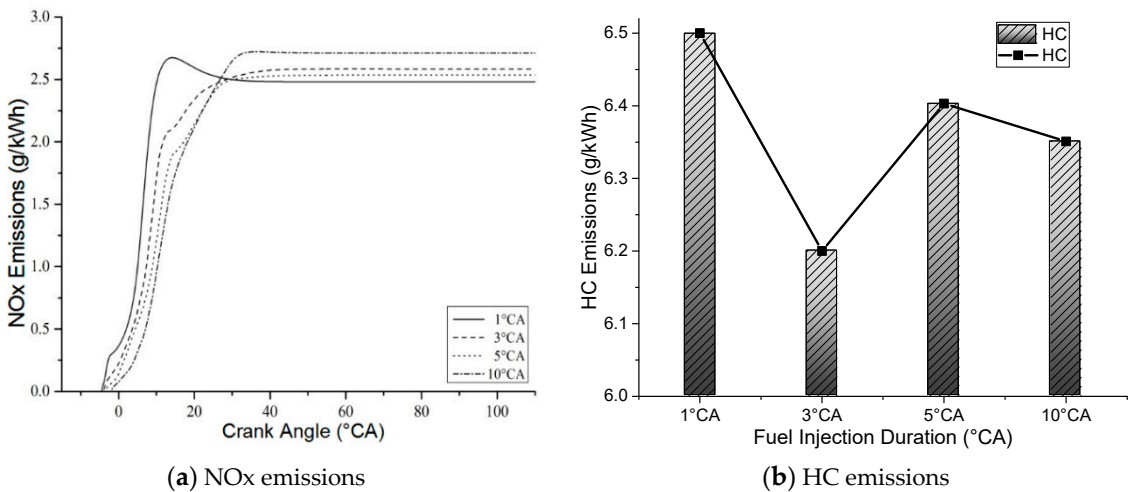

(**a**) NOx emissions   (**b**) HC emissions

**Figure 22.** NOx and HC emissions under different pilot injection durations.

## 4. Conclusions

In the present work, the pilot fuel injection parameters in PCC of a 2-stroke marine LP-DF engine was investigated using the Multidimensional CFD calculation. The main conclusions are listed as follows:

(1) The number of pilot injector orifices affected the ignition of pilot fuel and the flame propagation speed inside the combustion chamber. Due to the small space of the pre-combustion chamber, a large number of nozzle orifices caused the pilot oil beam to hit the wall of the pre-combustion chamber—affecting the atomization and evaporation of the oil droplets. When the number of pilot injector orifices was 1, engine performance and emission characteristics were very good.

(2) The SOI timing of pilot had a great impact on engine performance. With the delay of pilot injection timing, the in-cylinder pressure and the average temperature decreased significantly, the duration of heat release increased, and the engine performance deteriorated. In addition, NOx emissions gradually decreased with the delay of SOI timing of pilot.

(3) The duration of pilot injection mainly affected the ignition delay of pilot fuel and had a greater impact on the flame propagation. The longer the pilot injection duration, the later the pilot was ignited at the compression stroke and the lower the cylinder pressure and the average temperature were. The long duration of pilot injection resulted in poor engine performance and increased

NOx emissions. HC emissions showed a trend of first decreasing and then increasing with the increase of injection duration.

**Author Contributions:** H.G. and S.Z. contributed to the case study and the original manuscript.; J.Z. checked the results of the whole manuscript; M.S. translated the original manuscript. All authors have read and agreed to the published version of the manuscript.

**Funding:** The authors appreciate the support by a grant from National Natural Science Foundation of China (Grant No. U1906232) and the China financial support of Marine Low-Speed Engine Project-Phase I (Grant No. MC-201501-D01-03).

**Acknowledgments:** Great thanks for valuable comments and suggestions of the reviews and the editors of Processes.

**Conflicts of Interest:** The authors declare no conflict of interest.

## Abbreviations

| | |
|---|---|
| 1D | One-Dimensional |
| 3D | Three-Dimensional |
| aTDC | After Top Dead Center |
| bTDC | Before Top Dead Center |
| BSPC | Brake Specific Pilot Fuel Consumption |
| BSGC | Brake Specific Gas Consumption |
| °CA | Crank Angle Degree |
| CFD | Computational Fluid Dynamics |
| $CH_4$ | Methane |
| CO | Carbon Monoxide |
| ECAs | Emission Control Areas |
| EGR | Exhaust Gas Re-circulation |
| HC | Hydrocarbon |
| HCCI | Homogeneous Charge Compression Ignition |
| HFID | Heated Flame Ionization Detector |
| IMO | International Maritime Organization |
| LNG | Liquefied Natural Gas |
| LP-DF | Low-Pressure Dual-Fuel |
| LSFO | Low Sulphur Fuel Oil |
| MCC | Main Combustion Chamber |
| NG | Natural Gas |
| NOx | Nitrogen Oxides |
| PCC | Pre-combustion Chamber |
| ROHR | Rate of Heat Release |
| SCR | Selective Catalytic Reduction |
| SOI | Start of Injection |
| SOx | Sulfur Oxides |
| TDC | Top Dead Center |
| WinGD | Winterthur Gas and Diesel |

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
