# Peer review of "A Numerical Study on the Pilot Injection Conditions of a Marine 2-Stroke Lean-Burn Dual Fuel Engine"

_processes, doi:10.3390/pr8111396_

Round 1

Reviewer 1 Report

The manuscript describes modelling work of a marine 2-stroke LP-DF engine and the effect of pilot fuel injection. The topic is relevant for this journal, however, some issues should be resolved before its publication:

  • In the Introduction, it would be useful if you make clearer the novelty of the work and the main objectives.
  • Once the objectives/scope are better described, they should be referred to in the Conclusions section

Minor comments:

-In line 85, I think you mean 'The marine LP-DF engine is strong...'

-In line 237, the paragraph starts referring to Figure 10, I believe it's about Figure 12, please verify this.

Author Response

Response to Reviewer 1 Comments

Dear Reviewer,

Thank you very much for your comments. The manuscript is entitled as follows: “A Numerical Study on the Pilot Injection Conditions of a Marine 2-stroke Lean-burn Dual Fuel Engine” (Manuscript ID: processes-976054). We also wish to take this opportunity to thank the reviewers for their constructive comments and valuable recommendations. We have carefully revised the manuscript according to reviewers’ suggestion.

Our responses to the comments are listed below:

  • Point 1: In the Introduction, it would be useful if you make clearer the novelty of the work and the main objectives.
  • Once the objectives/scope are better described, they should be referred to in the Conclusions section

Response 1: Yes, we fixed the introduction of the last paragraph:

In a 2-stroke lean-burn dual-fuel engine, the occurrence of knock limits the increase in power. Therefore, the aim of this study is to improve the dual-fuel engine performance and to reduce the occurrence of knocking by optimizing pilot injection conditions. A three-dimensional (3D) model of the 2-stroke LP-DF engine was developed and validated with the help of CFD simulation software (Section 2). Then, the effects of pilot fuel injection conditions in PCC on LP-DF engine emissions and performance were studied from three aspects: injector orifices, pilot fuel SOI timing, and injection duration (Section 3). The main simulation results and conclusions were summarized at last (Section 4).

Point 2: Minor comments:

-In line 85, I think you mean 'The marine LP-DF engine is strong...'

-In line 237, the paragraph starts referring to Figure 10, I believe it's about Figure 12, please verify this.

Response 2: Yes, we fixed line 85 and line 237. The main modifications are as follows:

The marine LP-DF engine is different from the traditional diesel engine in the processes of air exchange, mixture formation, ignition, and flame propagation

Figure 12 shows the temperature distribution in the combustion chamber at different pilot fuel SOI timings.

Thank you very much for the excellent and professional revision of our manuscript.

Yours sincerely,

Hao Guo

Reviewer 2 Report

In this manuscript, the authors present a numerical study on the pilot injection conditions of a marine 2-stroke LB-DF engine. They used a 3D  CFD-model of a 2-stroke WinGD RT-Flex50DF engine as a model. Three aspects that are injection orifices, pilot fuel SOI timing, and injection duration are studied to investigate the effect of pilot fuel injection conditions in PCC on LP-DF engine emissions.

In my opinion, the manuscript is well written, covers a relevant topic, and it is easy to read. The research methods are described well, and the results are clearly presented.

I have only one note;

In line 117, Figure 1 has been used in [26 ] (Guo, H.; Zhou, S.; Shreka, M.; Feng, Y. Effect of Pre-Combustion Chamber Nozzle Parameters on the Performance of a Marine 2-Stroke Dual Fuel Engine. Processes 2019, 7, 876., it is 26th reference). Although it is used as a 3D model, it would be appropriate to cite [26] as the source.

I recommend to Editor In Chief to consider the paper for publication in Processes Journal.

Author Response

Response to Reviewer 2 Comments

Dear Reviewer,

Thank you very much for your comments. The manuscript is entitled as follows: “A Numerical Study on the Pilot Injection Conditions of a Marine 2-stroke Lean-burn Dual Fuel Engine” (Manuscript ID: processes-976054). We also wish to take this opportunity to thank the reviewers for their constructive comments and valuable recommendations. We have carefully revised the manuscript according to reviewers’ suggestion.

Our responses to the comments are listed below:

Point 1: I have only one note; In line 117, Figure 1 has been used in [26] (Guo, H.; Zhou, S.; Shreka, M.; Feng, Y. Effect of Pre-Combustion Chamber Nozzle Parameters on the Performance of a Marine 2-Stroke Dual Fuel Engine.Processes 2019, 7, 876., it is 26th reference). Although it is used as a 3D model, it would be appropriate to cite [26] as the source.

Response 1: Yes, we added the References:

Referring to the mean in-cylinder pressure in Figure 5(a), the error between the experiment and the CFD calculation results is less than 3.6% [28].

Thank you very much for the excellent and professional revision of our manuscript.

Yours sincerely,

Hao Guo

Reviewer 3 Report

The Authors correctly define the problem and run simulations. The strengths of the manuscript are: introduction with an overview of the state of the art and rationale for the research. Unfortunately, the readability of the manuscript is difficult due to the lack of key data for calculations: (elementary analysis of  natural gas, the same fuel was delivered to the engine and pilot burner?), and the poor description of the experimental investigations:

a) Which method was used to measure NOx, HC?

b)What was the excess air ratio, what was the gas temperature inlet?.

Main comments requiring clarification:

  1. Figure 5. The manuscript lacks information on the conditions of NOx measurement (flue gas temperature, measurement methods, what was measured NO, NO2?). It is also necessary to provide in this case how much oxygen was in the flue gas. No discussion of the results. How can you explain step changes in NOx concentrations in the engine load function (Fig 5b)
  2. Page 2: The Authors use: "CH4 equivalence ratio", and then on page 16, "Abbreviations ... λ Air-Fuel Ratio". Please correct.
  3. Page 16: “HC- hydrocarbon” Only one hydrocarbon measured /obtained from simulation? When describing the simulation results, it is absolutely necessary to specify which hydrocarbons were formed. Which hydrocarbon had the highest concentration?
  4. The abbreviations are not listed alphabetically - which makes it difficult to read.

Author Response

Response to Reviewer 3 Comments

Dear Reviewer,

Thank you very much for your comments. The manuscript is entitled as follows: “A Numerical Study on the Pilot Injection Conditions of a Marine 2-stroke Lean-burn Dual Fuel Engine” (Manuscript ID: processes-976054). We also wish to take this opportunity to thank the reviewers for their constructive comments and valuable recommendations. We have carefully revised the manuscript according to reviewers’ suggestion.

Our responses to the comments are listed below:

Point 1: a) Which method was used to measure NOx, HC?

b) What was the excess air ratio, what was the gas temperature inlet?

Figure 5. The manuscript lacks information on the conditions of NOx measurement (flue gas temperature, measurement methods, what was measured NO, NO2?). It is also necessary to provide in this case how much oxygen was in the flue gas. No discussion of the results. How can you explain step changes in NOx concentrations in the engine load function (Fig 5b)

Response 1: Really good comments. We provide more information about our experiment. The main modifications are as follows:

The atmospheric pressure in the engine experiment was 101.7 kPa and the scavenging air temperature was 302.1 K. Besides, the lower calorific value of the natural gas fuel was 47.64 MJ/kg and the natural gas contained 97.07 mol% CH4 and 2.93% other gases (0.24 mol% ethane, 0.01 mol% propane, etc.). The excess air ratio of the lean-burn dual-fuel engine was close to 2.2. Moreover, the oxygen content in the flue gas under 25%, 50%, 75%, and 100% engine loads were 13.59%, 14.21%, 14.26%, and 13.91%, respectively. Furthermore, heated flame ionization HC detector (HFID) and chemiluminescent NOx analyzers were used in the experiment.

Table 1 Properties of tested equipments

tested gas

Type and method

NOx

CAI 400S HCLD

CO

CAI 300 NDIR

CO2

CAI 300 NDIR

HC

CAI 300 HFID

Point 2: Page 2: The Authors use: "CH4 equivalence ratio", and then on page 16, "Abbreviations ... λ Air-Fuel Ratio". Please correct.

Response 2: Yes, we deleted the λ Air-Fuel Ratio.

Point 3: Page 16: “HC- hydrocarbon” Only one hydrocarbon measured /obtained from simulation? When describing the simulation results, it is absolutely necessary to specify which hydrocarbons were formed. Which hydrocarbon had the highest concentration?

Response 3: The main modifications are as follows:

Among the three simulation cases, engine performance and NOx emission characteristics are both better for the single injector orifice case. Besides, CH4 accounts for the majority of the HC emissions and had the highest concentration. Methane slip and unburned HC are the main causes of HC emissions from LP-DF engines. Since only the number of injector orifices is changed in the simulation cases, the mass of the direct methane slip remains unchanged. The results show that the combustion duration is longer and the combustion is more sufficient under 4 injector orifices, which leads to the reduction of unburned HC.

Figure 11. NOx and HC emissions with different injector orifices.

Point 4: The abbreviations are not listed alphabetically.

Response 4: The main modifications are as follows:

1D

One-Dimensional

3D

Three-Dimensional

aTDC

After Top Dead Center

bTDC

Before Top Dead Center

BSPC

Brake Specific Pilot Fuel Consumption

BSGC

Brake Specific Gas Consumption

°CA

Crank Angle Degree

CFD

Computational Fluid Dynamics

CH4

Methane

CO

Carbon Monoxide

ECAs

Emission Control Areas

EGR

Exhaust Gas Re-circulation

HC

Hydrocarbon

HFID

Heated Flame Ionization Detector

IMO

International Maritime Organization

LNG

Liquefied Natural Gas

LP-DF

Low-Pressure Dual-Fuel

LSFO

Low Sulphur Fuel Oil

MCC

Main Combustion Chamber

NG

Natural Gas

NOx

Nitrogen Oxides

PCC

Pre-Combustion Chamber

ROHR

Rate of Heat Release

SCR

Selective Catalytic Reduction

SOI

Start of Injection

SOx

Sulfur Oxides

TDC

Top Dead Center

WinGD

Winterthur Gas & Diesel

Thank you very much for the excellent and professional revision of our manuscript.

Yours sincerely,

Hao Guo

Round 2

Reviewer 3 Report

The Authors have revised the manuscript, so I recommend that it be published in Processes journal.